# Effect of Bacterial Cellulose Plasma Treatment on the Biological Activity of Ag Nanoparticles Deposited Using Magnetron Deposition

**DOI:** 10.3390/polym14183907

**Published:** 2022-09-19

**Authors:** Alexander Vasil’kov, Alexander Budnikov, Tatiana Gromovykh, Marina Pigaleva, Vera Sadykova, Natalia Arkharova, Alexander Naumkin

**Affiliations:** 1A. N. Nesmeyanov Institute of Organoelement Compounds, Russian Academy of Sciences, 28 Vavilov St., Moscow 119991, Russia; 2Department of Biotechnology, Moscow Polytechnic University, Bolshaya Semyonovskaya Str., 38, Moscow 107023, Russia; 3Faculty of Physics, Lomonosov Moscow State University, Leninskie Gory 1–2, Moscow 119991, Russia; 4G. F. Gause Institute of New Antibiotics, 11 Bol’shaya Pirogovskaya St., Moscow 119021, Russia; 5FSRC “Crystallography and Photonics” RAS, 59 Leninsky Prospekt, Moscow 119333, Russia

**Keywords:** bacterial cellulose, plasma treatment, magnetron sputtering, silver nanoparticles, antimicrobial activity, X-ray photoelectron spectroscopy

## Abstract

New functional medical materials with antibacterial activity based on biocompatible bacterial cellulose (BC) and Ag nanoparticles (Ag NPs) were obtained. Bacterial cellulose films were prepared by stationary liquid-phase cultivation of the *Gluconacetobacter hansenii* strain GH-1/2008 in Hestrin–Schramm medium with glucose as a carbon source. To functionalize the surface and immobilize Ag NPs deposited by magnetron sputtering, BC films were treated with low-pressure oxygen–nitrogen plasma. The composition and structure of the nanomaterials were studied using transmission (TEM) and scanning (SEM) electron microscopy and X-ray photoelectron spectroscopy (XPS). Using electron microscopy, it was shown that on the surface of the fibrils that make up the network of bacterial cellulose, Ag particles are stabilized in the form of aggregates 5–35 nm in size. The XPS C 1s spectra show that after the deposition of Ag NPs, the relative intensities of the C-OH and O-C-O bonds are significantly reduced. This may indicate the destruction of BC oxypyran rings and the oxidation of alcohol groups. In the Ag 3d_5/2_ spectrum, two states at 368.4 and 369.7 eV with relative intensities of 0.86 and 0.14 are distinguished, which are assigned to Ag^0^ state and Ag acetate, respectively. Nanocomposites based on plasma-treated BC and Ag nanoparticles deposited by magnetron sputtering (BCP-Ag) exhibited antimicrobial activity against *Aspergillus niger*, *S. aureus* and *Bacillus subtilis*.

## 1. Introduction

The use of green technologies for the production of medical materials can significantly reduce or eliminate their negative impact on the environment. Simultaneously, there is a steady trend towards the widespread use of renewable resources (cellulose, chitosan, etc.) for medical purposes. Dressings based on biodegradable bacterial cellulose are a good alternative to synthetic polymeric materials [1]. Indeed, sugar and molasses, the production of which is effectively established in the food industry, are used as raw materials for BC production [2,3]. Technologies for the production of BC fibers, tubes and films have been developed, which makes it possible to obtain nonwoven cover bandages for the treatment of wounds and burns [4,5].

Burned patients are especially at risk of bacterial infections due to local and systemic immune dysfunctions. Fungal infection risk factors (prolonged intensive-care-unit stay, mechanical ventilation, broad spectrum antibiotherapy, central venous lines, systemic antibiotics) are frequent in severe burn care and predispose the patient to potentially serious fungal wound infections, whether associated with bacterial infection or not [6]. Although filamentous fungal infections (FFIs) caused by *Aspergillus niger* are not frequent, this fungal infection can cause severe wound infections in patients with extensive burns. Moreover, *Aspergillus* seems to be related with the higher mortality [7,8]. 

However, BC itself does not have an antibacterial effect and cannot prevent a possible secondary bacterial infection of the skin. To impart antibacterial properties to BC, the material is impregnated with antibiotics [6] and/or with Ag, Cu, ZnO, or TiO_2_ nanoparticles [9,10,11,12,13,14]. The latter are considered more preferable because pathogenic organisms are rapidly developing resistance to antibiotics, which presents a huge problem for modern medicine [15]. 

Ag NPs/bacterial cellulose composites represent an active area of research for medical applications as evidenced by some recent reviews [16,17,18,19]. An important role in their preparation is played by the development of new environmentally friendly technologies. Some of them are based on physical approaches without introducing any chemical reagents. Plasma treatment, magnetron sputtering, hydrothermal synthesis, photoirradiation, etc., are widely used [20,21,22,23,24,25,26]. 

When AgNO_3_ is used as a source of silver, it is supposed to be reduced by bacterial cellulose. However, as a rule, there are no data on the degree of Ag reduction and elemental composition, which can be used to estimate the content of unreacted AgNO_3_, while pure silver is deposited during magnetron sputtering.

The modification of the polymer surface by low-temperature low-pressure cold plasma treatment using inert or active gases promotes the formation of functional groups on the surface that can stabilize metal nanoparticles. Plasma treatment changes the surface morphology, increasing the roughness of the material used, and also improves its biocompatibility with human tissues [27,28]. A significant number of works have been devoted to the effects of processing polymers for medical and food purposes with various types of plasma [29,30,31,32,33]. The mechanisms of modification of inert -CH_2_ groups for various polymers by their oxidation with active plasma particles and vacuum UV radiation were investigated [27]. However, the use of preliminary plasma treatment of the BC surface with mixtures of oxidizing gases to stabilize metal nanoparticles deposited by magnetron sputtering has been little studied [34]. Oxygen-nitrogen plasma treatment allows the surface of polymers to be modified to form active oxygen-containing groups through direct oxidation with active gases, or rearrangement of charged oxygen-containing groups on the surface into oxidized forms [35]. The interaction of nanoparticles of biologically active metals with such groups leads to their immobilization and effective chemical stabilization, which contributes to the manifestation of bactericidal properties. 

It is known that with an increase in the etching time, the surface roughness of fibrous polymers increases, numerous “sharp peaks” of considerable height appear and the surface area increases, which promotes the adsorption of nanoparticles. For etching polymers of different nature, the most effective and affordable method is the use of plasma of atmospheric composition in various instrumentations [36,37].

To activate cellulose against pathogenic microorganisms, nanoparticles of various metals and oxides were applied to it both by traditional chemical methods and by magnetron sputtering. The surface was modified by magnetron co-deposition of Ag and SiO_2_ nanoparticles without preliminary plasma treatment of cellulose [38]. The resulting material showed activity against *S. aureus*, *E. coli*, and *C. albicans* strains. Magnetron deposition allows one to control the process parameters and set the size and composition of nanoparticles. Compared to traditional methods using silver salts, plasma processing allows materials to be obtained free of toxic precursor residues [39].

In this work, new hybrid materials with antibacterial activity against pathogens have been prepared by plasma treatment of bacterial cellulose film (BCP) and subsequent deposition of Ag NPs by magnetron sputtering (BCP-Ag). The materials were studied by XPS, a surface-sensitive tool, which provides elemental and chemical compositions.

## 2. Materials and Methods

### 2.1. Production and Characterization of Bacterial Cellulose

To prepare bacterial cellulose, the *Gluconacetobacter hansenii* GH-1/2008 strain from the collection of VKPM B-10547 (Gause Institute of New Antibiotics, Moscow, Russia) was used. The strain is non-toxic and non-pathogenic to humans [40]. The used medium was composed of (g/L): sucrose (20.0), peptone (5.0), yeast extract (5.0), Na_2_HPO_4_ (2.7) and citric acid monohydrate (1.15). The inoculum was prepared by growing *G. hansenii* on this medium with the use of a rotary shaker ThermoStable IS-20 DAIHAN Scientific Co., Seoul, Korea at 30 °C for 3 days. Upon completion of the cultivation, the prepared bacterial cellulose films were separated from the culture broth via filtration, repeatedly washed with distilled water to remove the medium components, treated with 1.0 M NaOH solution at 80 °C for 2 h to remove cells and other impurities immobilized on the films, and finally washed with distilled water until a neutral pH of wash liquid was reached. The detailed preparation of the BC films was described elsewhere [41]. 

The degree of BC polymerization was determined by the intrinsic viscosity measurement of its solutions in cadoxen according to ASTM D1795-96 and ASTM D4243-99 standards [42]. The average value of polymerization degree was 900.

### 2.2. Modification of BC Surface by Plasma Treatment

For plasma treatment of the films, a modified high-voltage converter of a VUP-5 vacuum station (SELMI Ltd., Sumy, Ukraine) was used. It was connected to a gas-discharge glass chamber placed under a vacuum cap with a flat stainless steel electrode at the base and a flat aluminum electrode fixed in the upper part of the chamber. The scheme of the installation is shown in Figure 1.

The high-voltage discharge frequency was 15 ± 5 kHz, the source voltage was 3 kV, and the generator power was 100 W. The working zone was evacuated to a working pressure of 10 Pa, the composition of the active gas corresponded to atmospheric air. The samples were exposed to oxidizing plasma for 1, 3 and 5 min.

### 2.3. Metallization of BC Surface by Magnetron Sputtering

To modify BC with silver, a laboratory setup for DC magnetron sputtering built into the VUP-5 was used. A target made of compact silver (99.99%) 10 cm in diameter was mounted on the cathode, and BC samples were placed on the anode with the working side facing the target. The distance between the target and the sample was 40 mm. Argon (99.99%) was used as a source of bombardment gas ions. Before deposition, the target was degassed in vacuum for 5 min, until a base pressure of 5.0 × 10^−4^ Pa was reached. The metal was deposited at a pressure of 0.2 Pa, a voltage of 700 V, and a power of 400 W. The coating time was 60 s.

### 2.4. Scanning Electron Microscopy 

The surface morphology of the BC films was studied by low-voltage scanning electron microscopy (LVSEM) with a Scios (FEI, Waltham, MA, USA) microscope at an accelerating voltage of 1 kV. EDX studies were carried out using Oxford Instruments X-max EDX system¸ (Abingdon, Oxfordshire, UK).

### 2.5. Transmission Electron Microscopy 

The BCP-Ag samples were studied using Hitachi transmission electron microscope HT7700 (Tokyo, Japan). Images were acquired at 100 kV accelerating voltage. Before measurements the samples were mounted on a 3 mm copper grid with a carbon film and fixed in a grid holder.

### 2.6. X-ray Photoelectron Spectroscopy 

X-ray photoelectron spectra were acquired with an Axis Ultra DLD (Kratos, UK) spectrometer using monochromatized Al K*α* (1486.6 eV) radiation at an operating power of 150 W of the X-ray tube. Survey and high-resolution spectra of appropriate core levels were recorded at pass energies of 160 and 40 eV and with step sizes of 1 and 0.1 eV, respectively. Sample area of 300 μm × 700 μm contributed to the spectra. The samples were mounted on a sample holder with a two-sided adhesive tape, and the spectra were collected at room temperature. The base pressure in the analytical UHV chamber of the spectrometer during measurements did not exceed 10^−8^ torr. The energy scale of the spectrometer was calibrated to provide the following values for reference samples (i.e., metal surfaces freshly cleaned by ion bombardment): Au 4f_7/2_–83.96 eV, Cu 2p_3/2_–932.62 eV, Ag 3d_5/2_–368.21 eV. The electrostatic charging effects were compensated by using an electron neutralizer. Sample charging was corrected by referencing to the C-OH peak identified in the C ls spectrum to which a binding energy of 286.73 eV was assigned [43]. The background of inelastic electron energy losses was subtracted by the Shirley method. The elemental composition was calculated using atomic sensitivity factors included in the software of the spectrometer corrected for the transfer function of the instrument.

### 2.7. Antimicrobial Activity Assay

The antimicrobial activity of the BC films was measured by the disc-diffusion method. Disks of 150 μm thickness without nanoparticles (control) and coated with Ag NPs were used. Antifungal activity was assessed using test strain of yeast *C. albicans* ATCC 2091 and opportunistic filamentous fungi *Aspergillus niger* ATCC 16404. The spectrum of antibacterial activity was studied using test cultures of gram-positive bacteria strains—*Bacillus subtilis* ATCC 6633, *S. aureus* FDA 209 ATCC 6538 and gram-negative bacteria—*Escherichia coli* ATCC 25922 from the collection of cultures of the Gause Institute of New Antibiotics (Moscow, Russia). Bacteria and fungi were incubated at 37 °C for 24 h. Standard discs with antibiotics (amoxicillin—20 μg/disc, amphotericin B—40 μg/disc, manufactured by Institut Pasteur, (Saint-Petersburg, Russia) and sterile discs with BC without antibiotics were used as negative and positive controls.

## 3. Results and Discussion

The development of new effective and environmentally friendly methods for the synthesis of antibacterial drugs and materials is a research priority. An increase in the number of strains of microorganisms that are resistant to most antibiotics makes it worthless to apply previously developed and intensively used antibacterial drugs.

The use of metal nanoparticles, which have high biological activity and do not cause resistance of microorganisms, to obtain medical materials is one of the actively developing areas of research. Currently, silver nanoparticles are the preferred candidates for incorporation into medical polymeric materials to impart antibacterial properties to them [44]. As a rule, methods of the chemical reduction of metal salts are used to obtain a biopolymer matrix with embedded Ag NPs [45,46].

Nanoparticles can be prepared by reducing silver salt on the surface of BC with visible radiation [47]. In all cases, the researchers postulated a good antimicrobial activity of the obtained materials; however, the cytotoxic effect was not tested. These methods have a number of limitations, which significantly complicate the use of the obtained materials for biomedical purposes. These are the presence of a significant amount of impurities of surfactants and residues of synthesis products, as well as the difficulty of controlling the completeness of metal reduction [48]. It should also be noted that in some cases, significant thermal heating is required during the recovery process. This can lead to partial degradation of the biopolymer and, accordingly, to a change in its molar mass.

One of the promising methods for the synthesis of metal nanoparticles is magnetron sputtering, which makes it possible to apply Ag NPs of a given composition under controlled conditions. However, metal nanoparticles deposited onto the BC films are known to have poor adhesion to the surface [49]. To increase the adhesion properties of the surface and the ability to retain nanoparticles, polymers are exposed to various oxidizing agents [50]. 

In the literature, the processing time that is traditionally used for surface plasma treatment of the various polymer films and fibers is about 1–30 min at 10–100 W. A processing time of less than 1 min does not increase the hydrophilicity of cotton, cotton-PET and PP at a given power [51]. Our experiments reveal the same results for 1 min BC treatment, while an exposure time of more than 5 min significantly reduces the mechanical properties of the films. Taking into account the results obtained, samples treated with plasma for 3 min were examined. A mixture of gases of atmospheric composition was used as an active plasma, since pure oxygen burns out the BC surface, increasing the roughness, but not the number of oxidized groups required to stabilize deposited Ag NPs [52]. BC films were modified using combination of low-pressure low-temperature atmospheric plasma treatment and Ag magnetron sputtering. To assess the surface morphology of the prepared nanomaterials, SEM was implicated.

Figure 2 shows SEM images of the initial BC (a), treated with plasma (BCP) (b) and a nanocomposite covered with agglomerates of Ag nanoparticles (BCP-Ag) with different magnifications (c,d). Compared to the original BC, the plasma-treated polymer surface underwent significant changes. Its roughness and heterogeneity increased due to the destruction of the fibers. SEM analysis of BCP-Ag films showed a uniform distribution of silver particles over the surface of the plasma-treated polymer that can be seen on the microphotograph as a white network of dots (Figure 2c,d).

Figure 3 displays the distribution maps of C, O, Ag and the energy dispersive X-ray spectrum of the BCP-Ag nanocomposite. It can be concluded that silver present in the nanocomposite and is not uniformly distributed on the surface at this length scale. Elemental analysis through EDX confirmed the presence of silver on the coated fabric. 

It was shown that during magnetron deposition of metal nanoparticles on a plasma-modified surface, the surface morphology is intact [53]. Our results are same. Apparently, the distribution of nanoparticles over the nanocomposite surfaces is likely to be greatly influenced by the contour of inhomogeneities and oxidized regions formed during etching of BC, where they are stabilized. The SEM micrographs showed that Ag NPs deposited on the BC surface are located mainly in places with surface inhomogeneities—ridges, roughness, etc. 

This may be influenced by an uncompensated charge on these surface fragments, which leads to the coalescence of metal particles formed during magnetron evaporation. It was shown [54] that Ag NPs coalesce into worm-like structures on inert (not plasma-treated) surfaces upon prolonged deposition. When the surface is treated with plasma, this effect becomes even more pronounced. The particle pattern took the form of clusters based on partially destroyed networks of BC fibers.

In order to estimate the size of the deposited Ag NPs, a set of TEM images was obtained. Figure 4 shows a TEM micrograph of BCP-Ag nanocomposite fibers with the particle size distribution. Ag nanoparticles with sizes ranging from 5 to 35 nm were recorded. Larger particles represent aggregates in the form of a “bunch of grapes”, consisting of smaller particles. For statistical analysis hundreds of nanoparticles were taken into the account.

To evaluate chemical transformations of the BC surface induced by the plasma treatment, a comparative XPS study of untreated samples BC and BC-Ag (BC with deposited Ag) and treated for 3 min BCP and BCP-Ag was performed.

Figure 5 shows the C 1s photoelectron spectra, fitted with some Gaussian profiles using the reliable chemical shifts [43]. The peaks at ~285, 286.73, 288.06 and 288.6 eV correspond to the C-C/C-H, C-OH/C-O-C, O-C-O and C(O)O groups, respectively. The relative intensities of different groups and a characterization of the photoelectron peaks are presented in Table 1. It is clearly seen that plasma treatment and Ag deposition lead to their significant transformation. After modification of the original BC, an increase in relative intensity of C-C/C-H groups was observed. There is also an increase in the signal at 288.06 eV, which may be assigned to C=O groups formed as a result of the oxidation of BC by plasma. The relative intensities of C(O)O groups are rather similar. This may indicate the destruction of BC oxypyran rings and the oxidation of alcohol groups.

The elemental compositions of samples BC, BCP, BC-Ag and BCP-Ag obtained from high-resolution spectra atoms without taking into account impurity atoms are shown in Table 2. It shows significant decrease in surface oxygen content and O/C ratio during the treatment. 

Figure 6 shows the Ag 3d photoelectron spectra and MNN Auger spectra of the composites, and Table 3 shows their characteristics. The binding energies and the widths of the photoelectron peaks differ markedly, which indicate different chemical states of Ag atoms and the presence of at least two chemical states of atoms in BCP-Ag nanocomposite.

The presence of a plasmon loss peak at 372.3 eV in the spectrum of the BC-Ag sample (Figure 6a) indicates the Ag^0^ state [56], which is also confirmed by the value of 726.1 eV of the modified Auger parameter. The difference between the measured and literature values of the binding energies of photoelectron peaks, kinetic energies and Auger parameters should be attributed to both the size effect caused by the small size of Ag nanoparticles and the manifestation of local differential charging near Ag nanoparticles and the inhomogeneous distribution of Ag over the depth of the samples.

The use of the MNN Auger spectrum of the BC-Ag sample as a reference made it possible to discriminate two states in the Auger spectrum of the BCP-Ag sample, one of which is Ag^0^, and the second at kinetic energies of 349.7 and 351.2 eV should be assigned to Ag^δ+^ (Figure 6). To represent the spectrum as two states, the spectra were normalized by intensity in the region of high kinetic energies to achieve their best match, and the spectrum of the BC-Ag sample was subtracted from the Auger spectrum of the BCP-Ag sample, as shown in Figure 6. The area ratio of the Auger spectra BCP-Ag/BC-Ag is ~2.

Corresponding correlations are also observed in the kinetic energies and in the shape of Auger peaks. In the Ag 3d_5/2_ spectrum, two states at 368.4 and 369.7 eV with relative intensities of 0.86 and 0.14 are distinguished, which are assigned to Ag^0^ state and Ag acetate, respectively. The difference observed in the data obtained from the photoelectron and Auger spectra indicates a nonuniform distribution of the Ag^0^ state over the depth of the sample. This follows from the fact that the inelastic mean free path of photoelectrons is greater than that of Auger electrons in view of its dependence on kinetic energy. That is, in the case under consideration, Ag MNN Auger electrons are more surface-sensitive than Ag 3d photoelectrons. It is not strange that outer layers contain silver presumably in oxide form.

Figure 7 shows the O 1s spectra of the studied samples with selected chemical states of oxygen atoms, and Table 4 shows their characteristics. 

The O 1s spectrum of pristine cellulose is characterized with four peaks at 532.1, 532.8, 533.4 and 534.0 eV which are assigned to C(O*)O, C-OH, O-C-O and C(O)O* groups. The peaks at 532.8 and 533.4 with an intensity ratio of 3:2 are characteristic of cellulose. The peaks related to carboxylate groups have equal intensities. After the plasma treatment and deposition of Ag, a new peak at 531.7 eV related to C=O bond was recorded. In the O 1s spectra of the BC-Ag and BCP-Ag samples, an increase in the relative concentrations of C(O)O groups is observed in comparison with that of BC and BCP samples. As a result of plasma treatment of the original cellulose, a carbonyl group appears, while in the case of BC-Ag and BCP-Ag, the samples’ spectra are similar. In other words, the silver present protects the cellulose from potential damage. Therefore, we can conclude that it is mainly located in the upper layers and does not penetrate deep into the material. This is also confirmed by the quantification data (Table 4), from which it follows that the deposition of silver leads to a sharp decrease in the oxygen content. 

A significant decrease in the oxygen content in metallized samples and an increase in the carbon content can be explained by the screening effect of nanoparticle aggregates concentrated around oxygen-containing groups. In this case, the depths of secondary electrons escape for oxygen and carbon are different, for the O 1s photoelectrons it is smaller and can be screened to a certain extent by Ag. In addition, part of the oxygen can be bound by metal nanoparticles

The moderate antifungal activity of the obtained samples was investigated in relation to the test strain of opportunistic filamentous fungi *Aspergillus niger* ATCC 16404. After incubation, the inhibition zones for *Aspergillus niger* were found to be 8 ± 0.1 for BCP-Ag and 10 ± 0.4 mm for AmpB, respectively. Plasma treatment of the BC surface affects the antifungal activity of magnetron-deposited silver against the *A. niger* ATCC 16404 strain. Wang et al. [57] showed that this species easily degrades cellulose-containing materials in a humid environment, which causes certain difficulties when it was used for wound healing or product packaging. 

The antibacterial activity was studied using test cultures of Gram-positive bacteria strains—*Bacillus subtilis* ATCC 6633, *S. aureus* FDA 209 ATCC 6538—and Gram-negative bacteria—*Escherichia coli* ATCC 25922. 25922. Table 5 presents data on the antimicrobial activity of materials based on BC. The pristine BC and that treated with plasma showed no biological activity.

The BCP-Ag material exhibits moderate antibacterial activity against Gram-positive bacteria *B. subtilis* ATCC 6633 and weak activity towards *S. aureus* FDA 209 ATCC 6538 (Figure 8). The antimicrobial activities for BCP-Ag and BC-Ag towards *Bacillus subtilis* were equal, whereas BC-Ag was not active on *S. aureus*. However, no activity against Gram-negative bacteria was found for BC-based composites.

The zone of inhibition of the *A. niger* ATCC 16404 for the BCP-Ag composite is comparable to the amphotericin B-polyene macrocyclic antibiotic. Thus, the combination of BC plasma treatment and magnetron sputtering of metal can be used to impart new functional properties to the material.

## 4. Conclusions

By treating cellulose with low-frequency plasma and the simultaneous deposition of silver nanoparticles, a new material with antimicrobial properties was obtained. XPS analysis of the material surface made it possible to isolate various chemical groups and characterize the charge states of silver atoms using the Auger parameter. It was established that the combination of methods of plasma processing materials based on bacterial cellulose, synthesized by the producer *G. hansenii* GH-1/2008, with magnetron sputtering of Ag NPs, makes it possible to obtain medical and other products resistant to destruction by the mold fungi *Aspergillus niger* and possessing antibacterial activity against the Gram-positive bacteria *Bacillus subtilis* and *Staphylococcus aureus*.

XPS revealed that BC plasma treatment leads to the formation of new oxygen-containing functional groups on the polymer surface, which effectively stabilize Ag nanoparticles with a diameter of 5–35 nm. Ag nanoparticles exist on the surface of BC pre-treated by plasma as metal Ag and Ag acetate.

The films with plasma treatment and Ag nanoparticles demonstrated antimicrobial action against the Gram-positive bacteria *Bacillus subtilis* and *Staphylococcus aureus* and filamentous fungi. These findings might provide insights into the possible therapeutic application of BCP-Ag films as an agent to treat bacterial and fungal wound infections.

The proposed complex method for obtaining biologically active materials is an environmentally friendly technology that does not introduce contamination into the material, which is confirmed by XPS analysis at all stages of its production.

## Figures and Tables

**Figure 1 polymers-14-03907-f001:**
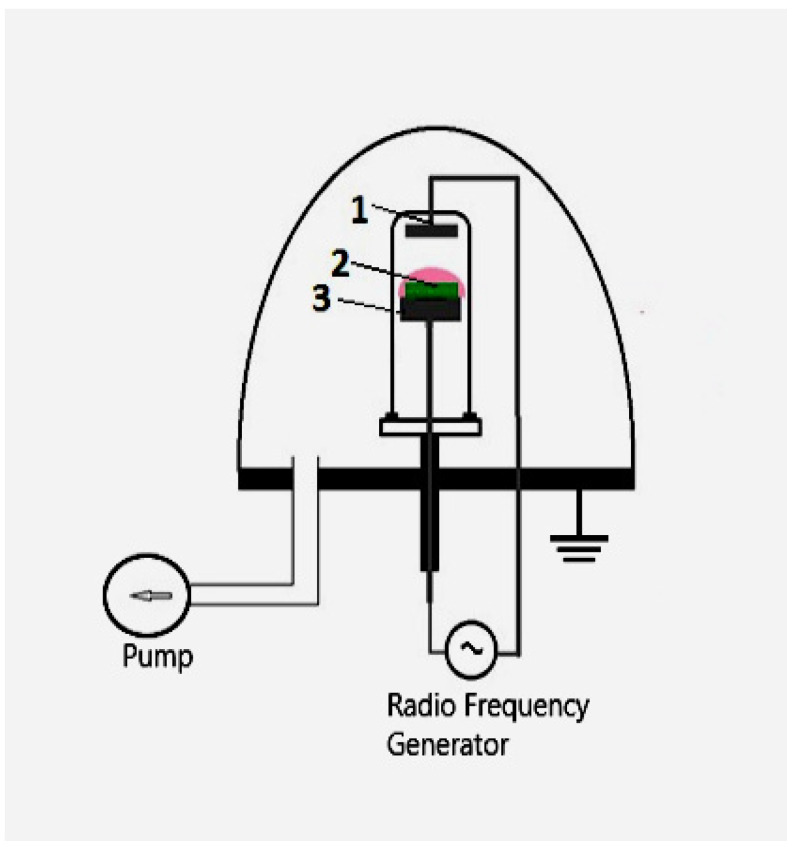
The installation for plasma treatment of the BC films: 1—an aluminum electrode; 2—a film placed on a grounded electrode; 3—a base of the grounded electrode.

**Figure 2 polymers-14-03907-f002:**
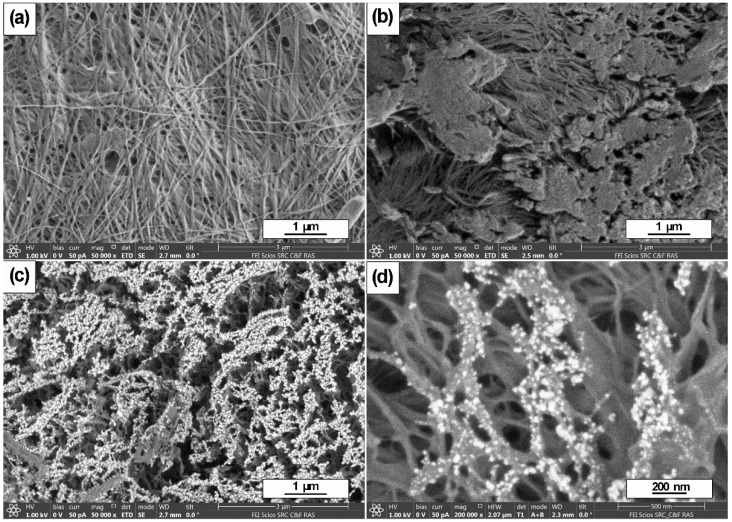
SEM images of the morphology of the initial bacterial cellulose film (BC) (**a**), BC after plasma treatment (BCP) (**b**) and BSE images of the BCP with agglomerates of Ag nanoparticles lying on its surface after Ag magnetron sputtering (**c**,**d**).

**Figure 3 polymers-14-03907-f003:**
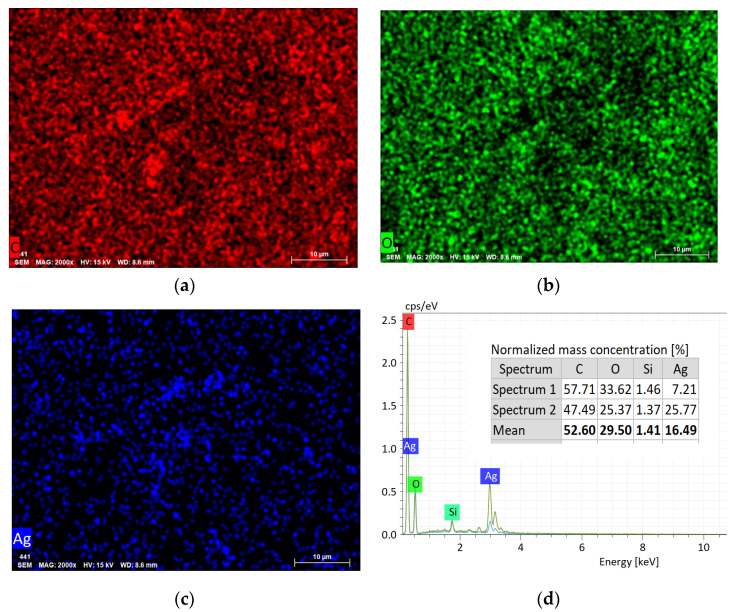
The elemental distribution of C (**a**), O (**b**), Ag (**c**) and energy dispersive X-ray spectrum (**d**) of the BCP-Ag nanocomposite.

**Figure 4 polymers-14-03907-f004:**
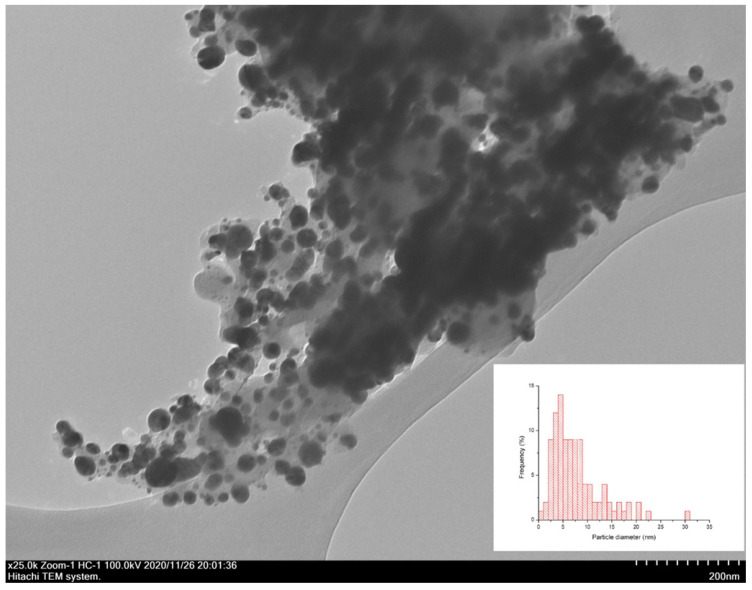
Bright field TEM image of the surface of BCP-Ag composite, obtained by plasma treatment of the BC and subsequent Ag nanoparticle deposition. In the right bottom corner is a histogram of the nanoparticles’ size distribution.

**Figure 5 polymers-14-03907-f005:**
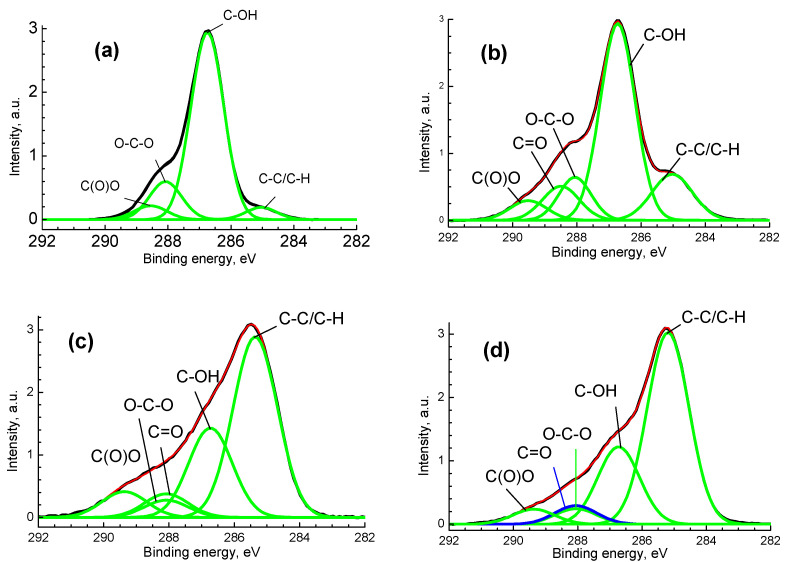
The С 1s spectra for pristine BC (**a**), BCP (**b**), BC-Ag (**c**) and BCP-Ag (**d**), black—experimental, red—fitting, green—individual peaks, blue—peak assigned to C=O group, which ovelapped with peak assigned to O-C-O group.

**Figure 6 polymers-14-03907-f006:**
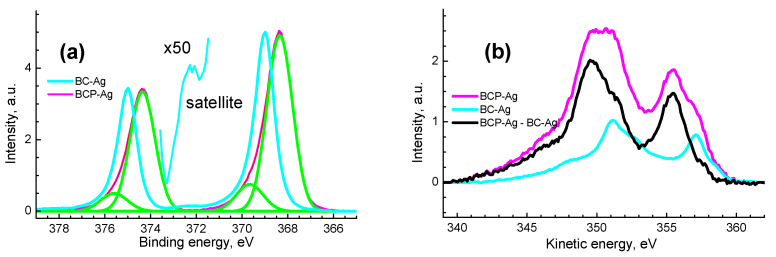
Ag 3d photoelectron spectra (**a**) and MNN Auger spectra (**b**) for BC-Ag and BCP-Ag nanocomposites.

**Figure 7 polymers-14-03907-f007:**
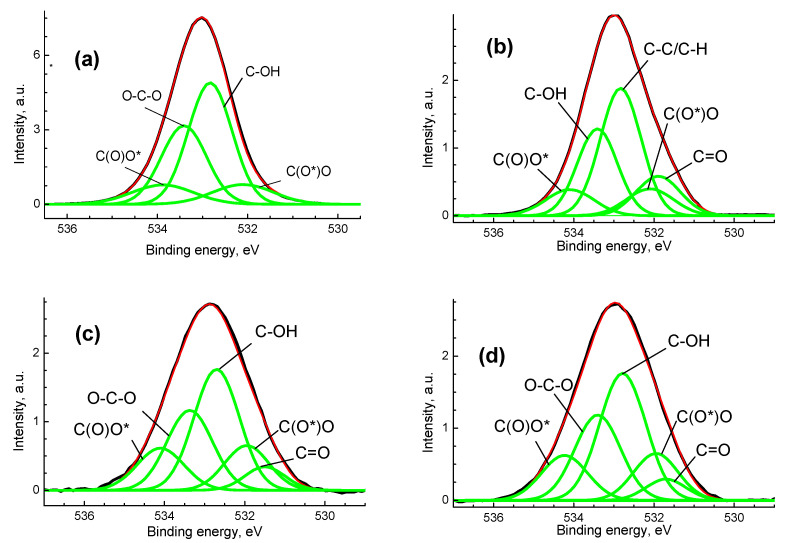
The O 1s photoelectron spectra for the BC (**a**), BCP (**b**), BC-Ag (**c**) and BCP-Ag (**d**), black—experimental, red—fitting, green—individual peaks.

**Figure 8 polymers-14-03907-f008:**
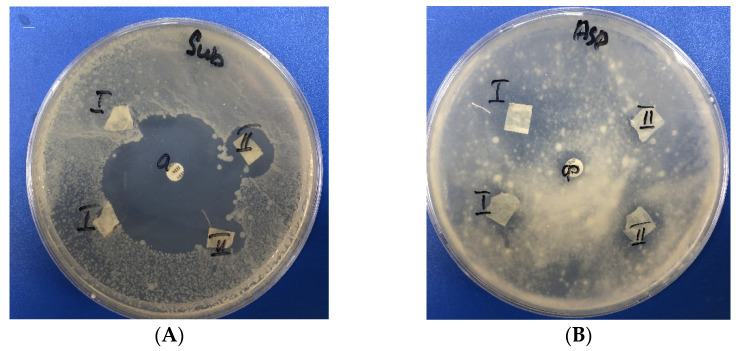
Antimicrobial activity of bacterial cellulose films with Ag and plasma treatment detected by disc-diffusion method: I—BC-Ag; II—BCP-Ag; (**A**) *B. subtilis* ATCC 6633; (**B**) *A. niger* ATCC 16404.

**Table 1 polymers-14-03907-t001:** Characteristics of the C 1s photoelectron spectra: binding energies (E_b_), Gaussian widths (W), and relative intensities (I_rel_) of photoelectron peaks belonging to different chemical groups.

Sample	Group	C-C/C-H	C-OH	O-C-O	C=O	C(O)O
BC	E_b_, eV	285.03	286.73	288.08		288.56
	W, eV	0.96	1.04	1.03		1.20
	I_rel_	0.05	0.75	0.15		0.05
BCP	E_b_, eV	285.03	286.73	288.06	288.5	289.51
	W, eV	1.26	1.09	1.0	1.15	1.15
	I_rel_	0.15	0.57	0.11	0.10	0.06
BC-Ag	E_b_,eV	285.4	286.73	288.06	288.06	289.37
	W, eV	1.36	1.34	1.34	1.34	1.37
	I_rel_	0.54	0.26	0.05	0.07	0.08
BCP-Ag	E_b_, eV	285.19	286.73	288.06	288.06	289.37
	W, eV	1.30	1.30	1.30	1.30	1.30
	I_rel_	0.60	0.24	0.05	0.06	0.05

**Table 2 polymers-14-03907-t002:** XPS quantification data (at. %) determined from the high-resolution spectra.

Samples	C	O	Ag	Ag/O	О/С	Ag/C
BC	42.3	57.7			1.36	
BCP	45.5	54.5			1.2	
BC-Ag	47.8	17.8	34.3	1.93	0.37	0.72
BCP-Ag	69.8	21.2	9.0	0.42	0.30	0.13

**Table 3 polymers-14-03907-t003:** Characteristics of photoelectron and Auger spectra for Ag and the nanocomposite: binding energies (E_b_), spin-orbit splitting (SOS), kinetic energies (E_k_) and Auger parameters.

	E_b_		SOS	E_k_		AP	
Sample	3d_5/2_, eV	3d_3/2_, eV	3d_3/2_- 3d_5/2_, eV	M_4_N_45_N_45_, eV	M_5_N_45_N_45_, eV	3d_5/2_+ M_4_N_45_N_45_, eV	State
Ag	368.327			357.855		726.182	Ag^0^ [55]
BC-Ag	369.0	375.0	6.0	357.1		726.1	Ag^0^
BCP-Ag	368.4	374.4	6.0	355.5	349.7	723.8	Ag^δ+^
	369.7	375.6	5.9	357.1	351.2	726.8	Ag^0^

**Table 4 polymers-14-03907-t004:** Characteristics of the O 1s photoelectron spectra: binding energies (E_b_), Gaussian widths (W) and relative intensities (I_rel_) of photoelectron peaks belonging to different chemical groups.

Sample	Group	C=O	C(O*)O	C-OH	O-C-O	C(O)O*
BC	E_b_, eV		532.1	532.8	533.4	534.0
	W, eV		1.00	0.91	0.91	1.15
	I_rel_		0.1	0.48	0.32	0.1
BCP	E_b_, eV	531.9	532.1	532.8	533.4	534.1
	W, eV	1.17	1.1	1.02	1.0	1.2
	I_rel_	0.13	0.1	0.4	0.27	0.1
BC-Ag	E_b_, eV	531.5	532.0	532.7	533.4	534.1
	W, eV	1.0	1.1	1.13	1.14	1.17
	I_rel_	0.07	0.14	0.39	0.26	0.14
BCP-Ag	E_b_, eV	531.7	532.0	532.8	533.4	534.2
	W, eV	1.00	1.10	1.15	1.14	1.15
	I_rel_	0.06	0.14	0.40	0.26	0.14

**Table 5 polymers-14-03907-t005:** Antimicrobial activity of the bacterial cellulose samples with Ag nanoparticle and plasma treatment.

Sample	Inhibition Zone, mm.				
	*Bacillus subtilis* ATCC 6633	*S. aureus* FDA 209 ATCC 6538	*Escherichia coli* ATCC 25922.	*Aspergillus niger* ATCC 16404	*C. albicans* ATCC 2091
BCP	0	0	0	0	0
BC-Ag	16 ± 0.4	0	0	0	0
BCP-Ag	18 ± 0.3	9 ± 0.1	0	8 ± 0.1	0
Amoxicillin 20 µg.	42 ± 0.7	27 ± 0.7	29 ± 0.2	not tested	not tested
Amphotericin B 40 µg.	not tested	not tested	not tested	10 ± 0.4	12 ± 0.6

## Data Availability

Not applicable.

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
