# Peer review of "Effect of Bacterial Cellulose Plasma Treatment on the Biological Activity of Ag Nanoparticles Deposited Using Magnetron Deposition"

_polymers, 2022, doi:10.3390/polym14183907_

Round 1
Reviewer 1 Report (Previous Reviewer 3)
Some of the suggestions I made in my first review have been overlooked by the authors. That said the manuscript can be accepted for publication.
Author Response
The authors thank the referee for a thorough analysis of the manuscript
Reviewer 2 Report (New Reviewer)
After carefully reviewing the article titled with "Effect of bacterial cellulose plasma treatment on the biological activity of Ag nanoparticles deposited using magnetron deposition", here there is a small suggestion that helps to the readers to understand the core idea of this manuscript.
Provide the better resolution image for Fig1.
In Fig.2d, the NP is not visible and the authors advised re capture the SEM images.
In antimicrobial resistance, please provide the deep data including zone of inhibition.
Author Response
We are grateful to the reviewers for a thorough analysis of our work. We are confident that referring to your comments has helped improve this manuscript.
After carefully reviewing the article titled with "Effect of bacterial cellulose plasma treatment on the biological activity of Ag nanoparticles deposited using magnetron deposition", here there is a small suggestion that helps to the readers to understand the core idea of this manuscript.
Comment: Provide the better resolution image for Fig1.
Response: The Fig. 1 has been changed.
Comment: In Fig.2d, the NP is not visible and the authors advised re capture the SEM images.
Response: Indeed, on the SEM image we can visualize agglomerates of Ag nanoparticles on BC. Individual nanoparticles can be registered using TEM due to the size of the nanoparticles (Figure 4). The caption and text have been changed.
Comment: In antimicrobial resistance, please provide the deep data including zone of inhibition.
Response: The text have been changed.
This manuscript is a resubmission of an earlier submission. The following is a list of the peer review reports and author responses from that submission.
Round 1
Reviewer 1 Report
In the paper "Effect of bacterial cellulose plasma treatment on the biological activity of Ag nanoparticles deposited using magnetron deposition" are presented interesting results on the fabrication of hybrid materials with antibacterial activity. The paper can be accepted for publication in Polymers mdpi after major revision. The following unclarities should be clarified.
I have no doubt the work has novelty but it is very important to highlight what is new in this work in comparison with papers published by authors previously.
It is not clear to me why nanoparticle aggregates concentrate around oxygen-containing groups and in places with surface inhomogeneities - ridges, roughness, etc. If I understood rightly the magnetron deposition should provide equal distribution of the silver nanoparticles on the bacterial cellulose. I think an appropriate discussion is important in this place.
It is very difficult for understanding where are the silver nanoparticles on Fig. 2. Please indicate the silver nanoparticles in Fig.2.
A caption for Fig.8 is difficult for understanding. Please rephrase.
Please cite relevant papers where similar new hybrid materials were fabricated:
https://doi.org/10.1016/j.colsurfa.2022.128525
https://doi.org/10.3390/ma13133037
English should be carefully checked.
Author Response
Пожалуйста, посмотрите приложение

Reviewer 2 Report
This work is about the functionalization of bacterial cellulose with AgNPs, is an important subject, but there is a lot of work done in this area, it is not clear what is the added value of this work and this work has many errors:
- there is a lot of confusion with the microorganisms used, throughout the text refer c. albicans ATCC 2091, but in table 5 there are no results for this microorganism, table 5 has results for s. aureus (missing the ATCC) is not mentioned in the abstract but referred to in conclusion. At "2.7 Antimicrobial activity assay" the authors refer amoxicillin and in the table 5 they have amoxiclav
-in table 5 what is the sample BC-Ag
- in figure 2 the authors use two detector modes: backscattered electron mode and secondary electron mode, why they use these two detectors and what results each one gives. In the figure 2 the images have different magnifications, so it is not possible to compare
- in the figure 3 the authors must present the quantification of the chemical elements, is Ag residual? if the Al is due to the sample holder, why do not they use another sample holder, in order to be sure of the existence or not of Al
- the authors should make the X ray mapping of Ag in the bacterial cellulose fibers to see the distribution of Ag.
- figure 4 shows a cluster of AgNPs, the AgNPs should be dispersed. The description of figure 8 has to be written, it is confusing
- the authors refer "The samples were exposed to oxidizing plasma for 1, 3 and 5 minutes" there is no discussion about this
- in the conclusion must be the results obtained in the work, the bibliographic references must be in the introduction.
Reviewer 3 Report
The manuscript titlted “Effect of bacterial cellulose plasma treatment on the biological activity of Ag nanoparticles deposited using magnetron deposition” present an hybrid bactericidal material prepared by plasma treatment of bacterial cellulose film (BCP) and subsequent deposition of Ag NPs by magnetron sputtering. The materials were studied by XPS, EDX, TEM and SEM.
The present work, although not of striking novelty, is rather interesting under the material engineeering point of view and of interest for the bactericidal material community. The manuscript is well organized, clearly written and to the point. That said, the english language needs to be checked and corrected for here and there.
The characterization, athough quite standard, is thoroughly performed and does provide relavant inside for applicative pourposes.
Some perspective on recent competing techniques for Ag NP encapsulation in bactericidal materials, other than magnetron sputtering, should be at least mentioned, see for instance “Antimicrobial Nanostructured Coatings: A Gas Phase Deposition and Magnetron Sputtering Perspective”, Materials 2020, 13(3), 784 (https://doi.org/10.3390/ma13030784) and references therein. This would widen the pletora of potential reader and update the bibliography to recent achievements in Ag NP encapsulaton.
For the above mentioned reasons I reccomend publication of the present manuscript after minor revisions as summarized in the following:
1. English revision required.
2. Inset of Figure 4 is hardly visible: replace it with a visible graph.
3. Figure 2 and 4: a length bar is required to apreciate the scale and should be inserted.
4. Do make a comment on the possibility of exploiting gas-phase techniques to obtain a simila materials: “Antimicrobial Nanostructured Coatings: A Gas Phase Deposition and Magnetron Sputtering Perspective”, Materials 2020, 13(3), 784 (https://doi.org/10.3390/ma13030784) and references therein.
5. Two key aspects in this field are: (a) the mechanical endurance of the material (b) the possible Ag NP uptake by the human body. The authors shoud at least briefly comment on these two aspects. I do not expect further investigation but at least a biref discussion/outlook of how thiese issues might affect their material, the discusson being based on existing recent litterature
Round 2
Reviewer 1 Report
The quality of the paper was essentially improved after revision. The paper can be accepted in its present form.
Reviewer 2 Report
The authors showed an effort to answer my questions/doubts, but it still has many errors, so I am of the opinion in rejecting the publication of this article